# One-Hot Encoding Strikes Back: Fully Orthogonal Coordinate-Aligned Class Representations

## Abstract

Representation learning via embeddings has become a central component in many machine learning tasks. This featurization process has gotten gradually less interpretable from each coordinating having a specific meaning (e.g., one-hot encodings) to learned distributed representations where meaning is entangled across all coordinates. In this paper, we provide a new mechanism that converts state-of-the-art embedded representations and carefully augments them to allocate some of the coordinates for specific meaning. We focus on applications in multi-class image processing applications, where our method Iterative Class Rectification (ICR) makes the representation of each class completely orthogonal, and then changes the basis to be on coordinate axes. This allows these representations to regain their long-lost interpretability, and demonstrating that classification accuracy is about the same or in some cases slightly improved.

## 1 Introduction

Embedded vector representations of structured data objects is nowadays a common intermediate goal for much of machine learning. The goal of these representations are typically to transform data into a form that is easy to work with for downstream applications, most centrally classification tasks. If the representations are successful, then for direct tasks only a simple classifier is required afterwards, e.g., logistic regression.

In this work, we argue that due to the fundamental nature of these representations, they should also aim for explicit interpretability. Note this is not attempting to make the process or neural architecture parameters used in arriving at these representations interpretable, but that given a data point's vector structure, one should be able to understand the components of its representation. In particular, we argue that for labeled classes provided among training data, that we should be able to (a) associate these classes with class mean vectors, (b) these class mean vectors should be completely orthogonal, and (c) each should be aligned with a particular coordinate (a one-hot encoding).

Given such an embedding of data point, then many tasks can be done directly by simply reading the representation. A multi-class classification task can be solved by returning the class associated with the coordinate with largest value. To understand a data point's relative association among multiple classes, one can compare their coordinate values; note that due to full orthogonality there are no hidden associations. If one fears there is implicit bias in a task, and that bias is associated with a captured class (e.g., gender bias captured by "woman" or "man" class), one can remove that class via projection like in Bolukbasi et al. (2016); Dev & Phillips (2019) – by simply not using those coordinates in downstream analysis. Other tasks without association to the bias should be unaffected, while those contaminated with bias will have that component removed.

A couple of recent papers have attempted to use neural networks to learn embedded representations which have class means orthogonal – their goal was increased generalization. The orthogonal projection loss (OPL) Ranasinghe et al. (2021), and CIDER Ming et al. (2023) both add a regularization term which favors compactness among points within a class and near orthogonality among class means. While these methods are useful seeding for our approach, we observe that they fail to produce class means that are nearly orthogonal. The average dot-product between normalized class means on CIFAR-100 is about $0.2$; for ours it is below $0.01$.

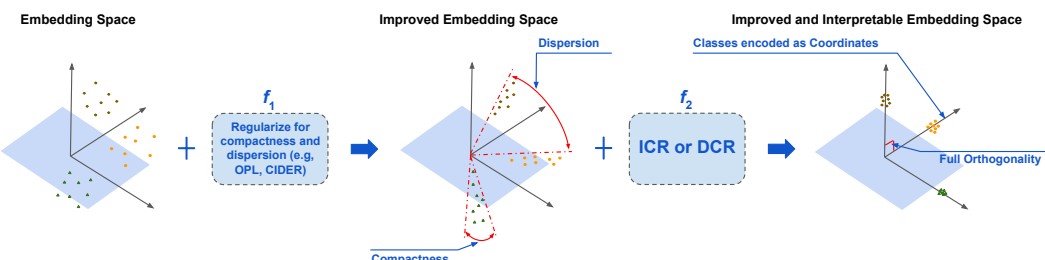

Figure 1: Our approach for embedding multi-class data: $f_1$ initializes classes to be clustered and dispersed. In $f_2$ our ICR and DCR make classes completely orthogonal, along coordinate axis.

Furthermore, our proposed framework structurally restricts the classifier to *classification-by-nearest-mean*, also known as the Rocchio algorithm. This directly reflects the training data: for each class, the mean of the training data is stored, and on evaluation of a test point, it is assigned a label of the nearest mean vector. This classification model produces a linear classifier with only 2 classes, and its standard evaluation reduces to the common task in information retrieval. This multi-class classifier becomes especially effective when the representation of the data is learned, and is common among state-of-the-art models (Yu et al., 2020) for few-shot learning approaches in image processing.

Our paper achieves the following:

1. We propose two class rectification methods (ICR and DCR) for multi-class classification under the Rocchio algorithm, which completely orthogonalize class means.

2. We prove that these methods either require one step (DCR), or iteratively converge to an orthogonal representation (ICR), conditioned that the class data is already clustered.

3. We show that this orthogonalized representation maintains state-of-the-art performance in a variety of classification tasks, given a backbone architecture.

The iterative class rectification (ICR) at the heart of our approach is an extension of a recent method ISR Aboagye et al. (2023) designed for bias reduction in natural language. That approach, ISR, required subspaces defined by two opposing classes (e.g., male-female, pleasant-unpleasant), which is restrictive. That paper only found a handful of such classes with sufficient training data, demonstrated the approach converged with two subspaces (2 pairs of classes), and did not always quickly converge to orthogonal on three subspaces (3 pairs of classes). A challenge addressed in that paper was determining a proper point of rotation. By using single-class means as we propose, this challenge goes away, and we show our approach effortlessly scales to 100 classes. We also introduce a second class rectification method (DCR) which achieves this result without iteration, but has less continuity.

After class means are fully orthogonal, we align them to coordinate axes. This basis transformation, by an orthogonal matrix, does not change any distance or dot-products between data representations.

**Example Uses.** Consider the CIFAR-100 test image with label orange; see Figure 2 and also Appendix E. The largest dot-products among the normalized class mean vectors for our technique (OPL+)ICR is with orange (0.995), the correct class, and then a big drop to cockroach at 0.0087 and other smaller values. In contrast the normalized class mean vectors for other approaches still identify orange as the correct class, but have much larger association with other classes. For OPL it is orange at 0.9975 but also apple, pear, and sweet_pepper between 0.82 and 0.72. Since the image is so associated with the class mean (dot product of virtually 1), we ascertain that the issue is the

| | top dot products for OPL | 0.9975 orange | 0.8204 apple | 0.7880 pear | 0.7215 sweet_pepper | 0.4562 poppy |
|---|---|---|---|---|---|---|
| | top dot products for OPL+ICR | 0.9950 orange | 0.0087 cockroach | 0.0061 maple_tree | 0.0059 girl | 0.0051 orchid |

Figure 2: Dot Products with class mean vectors for **orange** image with OPL and OPL+ICR.

| | top dot products for OPL | 0.8840 hamster | 0.7959 rabbit | 0.7698 mouse | 0.7161 squirrel | 0.6866 possum | 0.6296 fox |
|---|---|---|---|---|---|---|---|
| | top dot products for OPL+ICR | 0.7612 hamster | 0.4278 apple | 0.2296 pear | 0.1940 squirrel | 0.1847 kangaroo | 0.1323 baby |

Figure 3: Dot products with class mean vectors for **hamster+apple** image with OPL and OPL+ICR.

class means are not sufficiently orthogonal so that the image earns spurious correlation with the other classes. However, with ICR this does not happen since the class means are forced to be orthogonal.

Next, in Figure 3 we consider an image that has two classes present: hamster and apple. The representations vector has the largest dot-products with the normalized class means for OPL+ICR are 0.76 for hamster, 0.43 for apple, and the next largest has large drop to 0.23 for pear and 0.19 for squirrel. In contrast for OPL, the largest dot products are 0.88 for hamster, but then the next largest are for rabbit, mouse, squirrel, possum, and fox, all between 0.63 and 0.80. Because the hamster class has correlation with the other small fury mammals under OPL, they obscure the association with hamster and hide the association with apple which has a score of 0.58. This is not the case with OPL+ICR, so the association with pear and squirrel can be interpreted to geometrically represent uncertainty about those class labels.

Then we can consider removing the "hamster" class via a projection based approach (e.g., (Dev & Phillips, 2019)). Under OPL+ICR the largest dot-product is now apple, still at 0.43 and next largest are unchanged with pear (0.23) and squirrel (0.19). For OPL after projection, we also have the largest dot-product is with apple at 0.56, but somewhat obscured with other correlated classes including pear, orange, and sweet_pepper all between 0.45 and 0.52. Notably the other small fury mammals are also removed from strong association because of their correlation with the hamster class.

## 2    ALGORITHMIC FRAMEWORK

Our method considers a data set $Z \subset \mathcal{Z}$, where each $z_i \in Z$ is associated with a label $y_i \in [k]$, where $k$ is the number of distinct classes. We use image data $\mathcal{Z}$ as an exemplar. Then it operates in two phases towards creating an embedding in $\mathbb{R}^d$, with $d > k$; see Figure 1. The first phase learns an embedding $f_1 : \mathcal{Z} \to \mathbb{R}^d$ with the goal of classes being (nearly) linearly separable in $\mathbb{R}^d$. The second phase, the innovation of this paper, is another map $f_2 : \mathbb{R}^d \to \mathbb{R}^d$ which aims to retain (and perhaps improve) linear separability, but also achieve a form of orthogonality among classes. While this second phase can be interpreted as a form of learning–so it only sees training and not testing data–it is deterministic and does not follow the traditional *optimize parameters over a loss function*.

For input data $(Z, y)$, denote $X' = \{x'_i = f_1(z_i) \in \mathbb{R}^d \mid z_i \in Z\}$ as the embedding after phase 1. Then denote $X = \{x_i = f_2(x'_i) \in \mathbb{R}^d \mid x'_i \in X'\}$ as the embedding after phase 2. Let $Z_j$, $X'_j$, and $X_j$ be the data points in class $j \in [k]$ for the initial data, first, and final embedding, respectively.

**Rocchio algorithm.**    We leverage the Rocchio algorithm to build classifiers. For an embedded data set $(X, y)$, it first creates class means $v_j = \frac{1}{|X_j|} \sum_{x_i \in X_j} x_i$ for each class $j \in [k]$. Then on a training data point $x \in \mathbb{R}^d$ it predicts class $\hat{j} = \arg \min_{j \in [k]} \mathrm{D}(x, v_j)$. If we normalize all class means (so $v_j \leftarrow v_j / \|v_j\|$) then using Euclidean $\mathrm{D}(x, v_j) = \|x - v_j\|$ has the same ordering as cosine distance. That is, we can instead use $\hat{j} = \arg \max_{j \in [k]} \langle x, v_j \rangle$; we do this hereafter unless stated otherwise.

**Phase 1 embedding.**    For the first phase embeddings $f_1$ we leverage existing recent algorithms that aim for an embedding with three goals: (a) *accuracy*: each class can be (nearly) linearly separable from all other classes. (b) *compactness*: each class $X'_j$ has points close to each other, i.e., small variance. (c) *dispersion*: each pair of classes $j$ and $j'$ are separated, and in fact nearly orthogonal. In particular, a couple of recent papers proposed loss functions for $f_1$ as $\mathcal{L}_{f_1} = \mathcal{L}_{CE} + \lambda(\mathcal{L}_{comp} + \mathcal{L}_{disp})$. The $\mathcal{L}_{CE}$ is the standard cross entropy loss which optimizes (a), $\lambda \in [0, 1]$, and where $\mathcal{L}_{comp}$ and $\mathcal{L}_{disp}$ optimize (b) and (c). These are actualized with $|Z| = n$, $k$ classes, $n_1 = \sum_{j \in [k]} |Z_j|(|Z_j| - 1)$ and $n_2 = \sum_{j \in [k]} |Z_j|(n - |Z_j|)$ as:

$$\mathcal{L}_{comp} = 1 - \frac{1}{n_1} \sum_{j \in [k]} \sum_{z_i, z_{i'} \in Z_j} \langle f_1(z_i), f_1(z_{i'}) \rangle, \qquad \mathcal{L}_{disp} = \left| \frac{1}{n_2} \sum_{z_i \in Z_j; z_{i'} \in Z_{j' \neq j}} \langle f_1(z_i), f_1(z_{i'}) \rangle \right| (1)$$

$$\mathcal{L}_{comp} = -\frac{1}{n} \sum_{i=1}^{n} \log \frac{\exp(\langle f_1(z_i), v_{j_i} \rangle)}{\sum_{j=1}^{k} \exp(\langle f_1(z_i), v_j \rangle)}, \qquad \mathcal{L}_{disp} = \frac{1}{k} \sum_{j \in [k]} \log \frac{1}{k-1} \sum_{j' \neq j} \exp(\langle v_j, v_{j'} \rangle)(2)$$

The loss for OPL Ranasinghe et al. (2021) is in eq 1 and for CIDER Ming et al. (2023) in eq 2.

We observe (see Table 1), that these achieve good clustering among classes, but the classes are not fully orthogonal. On training data for CIFAR-100, they achieve about 98% accuracy or better. This holds under the trained linear classifier (under logistic regression) or the Rocchio algorithm. Pre-processing in phase 1 will prove an important first step for the success of our phase 2.

**Phase 2 embedding: Full orthogonalization.** As we observe that the result of *learning* an orthogonal embedding through regularization is not completely effective, the second phase provides a deterministic approach that *enforces* orthogonality of the class means. A first, but unworkable, thought is to *just run Gram-Schmidt* on the class mean vectors $v_1, \ldots, v_k$. However, this does not instruct a generic function $f_2$ that also applies to training data; if we recalculate their class means they are not orthogonal – our goal is that they are. Towards this end, we propose two approaches:

---

**Algorithm 1** BinaryICR($X, X_1, X_2$, iters: $T$)

1: **for** $i = 0, 1, \ldots, T-1$ **do**
2: $\quad v_1, v_2 \leftarrow$ normalized means($X_1, X_2$)
3: $\quad$ BinaryCR($X, v_1, v_2$)

---

**Algorithm 2** BinaryCR($X, u, v$)

1: Set $v' = v - \langle u, v \rangle u$
2: Define projection $\pi(\cdot) = (\langle \cdot, u \rangle, \langle \cdot, v' \rangle)$
3: **for** $x \in X$ **do**
4: $\quad \tilde{x} \leftarrow$ GradedRotat($\pi(u), \pi(v), \pi(x)$)
5: $\quad x \leftarrow x + (\langle \pi(u), \tilde{x} - \pi(x) \rangle u$
$\qquad\qquad + \langle \pi(v'), \tilde{x} - \pi(x) \rangle v')$

---

**Algorithm 3** BinaryDCR($X, X_1, X_2$)

1: $v_1, v_2 \leftarrow$ normalized means($X_1, X_2$)
2: $\theta' \leftarrow$ angle between $v_1$ and $v_2$; $\theta = \frac{\pi}{2} - \theta'$
3: **if** $(\theta' \leq \frac{\pi}{2})$ **then** set angle $\phi = \theta'/2$
$\qquad\qquad$ **else** set angle $\phi = \frac{\pi}{4}$
4: **for** $x \in \{x \in X \mid \langle v_2, x \rangle \leq \phi\}$ **do**
5: $\quad x \leftarrow R_\theta x$

---

Iterative Class Rectification (ICR): We adapt a recent approach called Iterative Subspace Rectification (ISR) (Aboagye et al., 2023) designed to orthogonalize language subspaces to reduce bias. This approach handles two concepts, each defined by a pair of classes (e.g., male-female, pleasant-unpleasant) as vectors $v_1, v_2$; and centers the data around these. Then it applies a "graded rotation" operation (Dev et al., 2021) (see Algorithm 5 in appendix) on the span of the two linear concept directions (span($v_1, v_2$)). Because it operates only in this span, it only alters the embedding in this 2-dimensional span. The graded rotation moves $v_2 \mapsto v_2'$ so it is orthogonal to $v_1$, and it applies a different rotation for each data, depending on the angles to $v_1, v_2$ so the amount of rotation continuously interpolates between that for $v_2 \mapsto v_2'$ and no rotation at $v_1$. The ISR paper (Aboagye et al., 2023) demonstrated that by *repeating* this process we get $v_2 \mapsto v_2^\star$, with $v_2^\star$ orthogonal to $v_1$, *even recomputing $v_1$ and $v_2^\star$* from the updated embedded data points which define the associated classes.

We adapt this process in two ways in this paper, in Algorithms 1 and 2. First we only use individual classes, and their class-means (line 2 of Alg 1), in place of concepts which spanned across two opposing ideas (and hence two sets of embedded points for each concept). Second, because we initialize with clustered concepts *by cosine similarity* around their class mean vectors, we can rotate around the origin (line 4 of Alg 2), and do not require a centering step as in ISR. Algorithm 2 does the core operation of projecting to the span of two subspaces $u, v$, apply GradedRotation on each point $x \in X$, and then adjust only the coordinates in span($u, v$) (line 5). Algorithm 1 iterates this procedure $T$ steps as the recalculated class means become orthogonal.

To apply this to all classes, we now do apply a Gram-Schmidt sort of procedure; see details in Algorithm 4. We first identify the class mean vectors most orthogonal (line 3), and apply one step of Binary ICR. Then at each round, we find and maintain the subspace of the class means we have

attended to so far $S_{j-1}$, and find the class mean $v_j$ most orthogonal to that subspace (line 8). We project $v_j$ onto $S_{j-1}$, to get $\bar{v}_j$, and then run one step of BinaryCR to orthogonalize $v_j$ from $\bar{v}_j$ (and hence to all of $S_{j-1}$). Once we have addressed all classes, we iterate this entire procedure a few times (typically $T = 1$ or 2 iterations, and not more than 5).

Finally, at the conclusion of the MultiClass ICR, the class means on the embeddings $v_1, \ldots, v_k$ are all orthogonal (up to several digits of precision). To complete the definition of function $f_2$, we add a final transformation step that aligns $v_1, \ldots, v_k$ to the first $k$ coordinate axes. This step is defined by a single orthogonal matrix, so it does not change Euclidean distance or dot products.

---

**Algorithm 4** MultiICR$(X, X_1, \ldots, X_k,$ iters: $T)$

---

1: **for** $i = 0, 1, \ldots, T - 1$ **do**
2:     Let $v_i$ be the normalized mean vector of class $X_i$ for $i = 1, 2, \ldots, k$.
3:     Set $r, s = \arg\min_{1 \le i, j \le k} |\langle v_i, v_j \rangle|$, WLOG suppose $r = 1$, $s = 2$
4:     Let $S_1$ and $S_2$ be the span of $\{v_1\}$ and $\{v_1, v_2\}$ respectively
5:     Run BinaryCR$(X, v_1, v_2)$
6:     Recalculate normalized class means $v_i$ for all $i$
7:     **for** $i = 3, \ldots, k$ **do**
8:         Choose $t = \arg\min_{j \ge i} \langle v_1, v_j \rangle^2 + \langle v_2, v_j \rangle^2 + \cdots \langle v_{i-1}, v_j \rangle^2$
9:         WLOG assume $t = i$
10:        Let $\bar{v}_i$ be the projection of $v_i$ onto $S_{i-1}$
11:        Set $u_i = v_i - \sum_{j=1}^{i-1} \langle v_j, v_i \rangle v_j$ and $v_i' = u_i / \|u_i\|$
12:        Run BinaryCR$(X, v_i', \bar{v}_i)$
13:        Set $S_i$ to be the span of $\{S_{i-1}, v_i\}$
14:        Recalculate class normalized means $v_j$ for all $j$

---

Discontinuous Class Rectification (DCR): This approach is similar, but does not require iteration, at the expense of a discontinuous operation. It replaces the graded rotation Dev et al. (2021) with a step that identifies a conical region around $v_2$, and applies an angle $\phi$ to all points in this region so afterwards $\langle v_1, v_2 \rangle = 0$. If the angle between $v_1$ and $v_2$ is acute, then the conical region is defined in the span of $v_1, v_2$ by an angle $\theta$ from $v_2$ to the bisector direction between $v_1$ and $v_2$. That is, for points closer to $v_2$, they are moved along with $v_2$, the rest are left alone. If $v_1$ and $v_2$ have an obtuse angle, then we set the conical angle around $v_2$ at $\pi/4$, so we only move points which will be closer to $v_2$ *after* the transformation when $\langle v_1, v_2 \rangle = 0$. The multiclass version of DCR follows the Gram-Schmidt recipe of ICR, but with no iteration.

**Freezing learned embedding $X'$.** It is important to note that before ICR or DCR is applied to determine $X$, we need to learn and *freeze* the initial embedding $X' \leftarrow f_1(Z)$. Then $f_2$ operates on $X'$, to create $X \leftarrow f_2(X')$ without adjusting $f_1$. There are slight differences in how OPL Ranasinghe et al. (2021) and CIDER Ming et al. (2023) choose an embedding layer: for OPL it is the penultimate layer, where as for CIDER it is the "head," the last layer. We follow recommendations in those works.

In evaluation mode, we also need a classifier. In Section 3.2, we discuss two ways to train classifiers – one is the Rocchio classifier (which we recommend for its structural properties, and since it needs no further training). However a common approach is to build a logistic regression model on the last layer of $f_2(f_1(Z))$; we also do this on the training data. Finally, we can consider the evaluation/test data $z \in Z_{\text{test}}$, which are evaluated with the chosen classifier operating on $f_2(f_1(z))$.

## 2.1 PROPERTIES OF ICR AND DCR

We highlight key properties of the ICR and DCR procedures. Proofs are deferred to Appendix A.

First, we show that binary ICR, even if iterated, only affects coordinates of data points in the span of the original mean vectors. This implies that the mean vectors of classes stay in their original span. Moreover, it implies that as MultiICR gradually includes more classes, it maintains a modified span, and all components of coordinates outside those spans are unchanged. Hence, if $d > k$, then the null space of the classes is unchanged under the MultiICR procedure. These results follow trivially for binaryDCR and MultiDCR since we just apply Gram-Schmidt procedure on class cones (the cones around the class means that contain the whole associated class).

Second, we show that this process converges to have the mean vectors completely orthogonal to each other. This argument requires that initial classes $X'_j$ are clustered, this explains and justifies the use of optimizing $f_1$ under the OPL or CIDER loss, or something similar, before applying BinaryICR. The assumption we use (Assumption 1; see also Appendix A.2) is probably more restrictive than necessary (it requires clusters are completely separable), but it makes already messy proofs manageable.

**Assumption 1** *Let $v_i$ be the mean of $X_i$, and let $X_i$ be included in the cone of radius $\phi_i$ around $v_i$ for $i = 1, 2, \ldots, k$. Assume theses cones are disjoint (except at the origin).*

**Theorem 1 (Convergence of BinaryICR)** *Let Assumption 1 hold with $k = 2$, and the angle between $v_1$ and $v_2$ is less than $\frac{\pi}{2}$. Then the BinaryICR algorithm converges: as we iterate, in the limit, the angle between class means approaches $\frac{\pi}{2}$.*

The comparable arguments for DCR are more straightforward. Following Assumption 1, all points of $X'_2$ are in a cone, and all of them and only them are updated in the operation. Since those points are all moved an angle exactly $\phi$, and $\phi$ moves $v_2$ orthogonal to $v_1$, then if we recalculate $v_2$ after the move, it will still be orthogonal to $v_1$. Hence this achieves the orthogonality goal after one step, and only affects data in the span of $v_1, v_2$.

**Theorem 2 (Convergence of BinaryDCR)** *Assume Assumption 1 holds with $k = 2$. In addition, if the angle between $v_1$ and $v_2$ is bigger than $\frac{\pi}{2}$, then we assume $\phi_1, \phi_2$ are less than $\frac{\pi}{4}$. Then after running the BinaryDCR algorithm, the class means will be orthogonal to each other.*

However, data may not be completely separable; we observe experimentally that OPL and CIDER achieve 99-100% accuracy in P@1 on the training data; see Appendix C. So instead, we can consider a robust version of $v_1, v_2$: set points in the appropriate conical regions as "inliers" and redefine robust $\bar{v}_1$ and $\bar{v}_2$ as the means of these inlier points only, then apply the DCR step, and then the $\bar{v}_1$ and $\bar{v}_2$ of the inliers will be orthogonal after one step. We observed that the difference in output from the original and robust version is in the third digit of precision, so only show results for the simpler non-robust variant of DCR.

The MultiDCR algorithm is running Gram-Schmidt algorithm on class cones such that normalized class means will constitute an orthonormal bases for a $k$-dimensional subspace of $\mathbb{R}^d$.

**Theorem 3 (Convergence of MultiDCR)** *Let Assumption 1 hold. In addition suppose that cones are sufficiently well-separated (see Assumption 3 in Appendix A.3). Then after running the MultiDCR algorithm, all class means will be orthogonal to each other.*

## 3 EXPERIMENTS

We evaluate our methods ICR and DCR in two main ways. First we show that these approaches, with high precision, achieve orthogonality of class means while previous approaches do not, and while maintaining good class compactness. Second, we show these approaches maintain or improve upon the near state-of-the-art accuracy in various learning frameworks. Note that ICR and DCR are designed to *maintain* class cohesiveness, not improve upon it, so we do not expect improvement on training data, and any improvement on the evaluation sets can be seen as a fortuitous effect of regularizing to a meaningful structure. We use standard image classification data sets and tasks.

**Datasets and Training Details.** In our main experiments, we use Resnet-9 as the backbone architecture for CIFAR-100 Krizhevsky (2009) classification task, and train for 120 epochs. The CIFAR-100 is an image dataset that consists of 60,000 natural images that are distributed across 100 classes with 600 images per class. All training, including ICR & DCR is performed on the training samples of 50,000 images. All evaluation is shown on the test data of the remaining 10,000 images.

### 3.1 ORTHOGONALITY AND COMPACTNESS

The dimension of the penultimate layer in OPL Ranasinghe et al. (2021) that was optimized towards being orthogonal was set to $d = 64$ dimensions. It is mathematically impossible to fit $k$ classes orthogonally for $k > d$ dimensions; note $k = 100$ for CIFAR-100 has $100 = k > d = 64$.

Alternatively, CIDER Ming et al. (2023) uses $d = 512$ dimensions in the final layer where dispersion and compactness is optimized. To help identify the best choice of $d$ we first measure geometric properties for OPL and CIDER for $d = 64, 128, 256, 512, 1024$. Table 1 shows for each: first, the average absolute dot-product between class means $\frac{1}{\binom{k}{2}} \sum_{j \neq j'} |\langle v_j, v_{j'} \rangle|$, and second, the average intra-class dispersion $\frac{1}{k} \sum_{j \in [k]} \frac{1}{X'_j} \sum_{x \in X'_j} \langle v_j, x \rangle$. For both, orthogonality increases (average dot products decrease) with higher dimensions, and while OPL's compactness keeps increasing, CIDER's decreases after $d = 128$. Notably, even at $d = 1024$, both OPL and CIDER are still far from orthogonal with an average dot-product of about $0.1$.

Table 1: Average class absolute dot products;                     and intra-class compactness.

| dim: | 64 | 128 | 256 | 512 | 1024 | 64 | 128 | 256 | 512 | 1024 |
|------|------|------|------|------|------|------|------|------|------|------|
| OPL | 0.2709 | 0.2412 | 0.1945 | 0.1509 | 0.1267 | 0.9742 | 0.9784 | 0.9851 | 0.9885 | 0.9900 |
| CIDER | 0.1602 | 0.1435 | 0.1247 | 0.1017 | 0.0930 | 0.9816 | 0.9809 | 0.9789 | 0.9764 | 0.9754 |

Next, in Table 2 we show the top-1 and top-5 accuracy for OPL and CIDER by dimension, on the CIFAR-100 evaluation set. OPL performs better than CIDER, and has the best top-1 accuracy at 1024 dimensions. Somewhat surprisingly, all others peak at smaller dimension (128 or 256), but the decrease is mostly not too significant. We decide to continue with the best result for top-1 accuracy and orthogonality, and so set $d = 1024$ dimensions as a default.

In Figure 4 we also plot block matrices for the absolute value of dot products between all pairs of class means, for OPL and CIDER embeddings at 64 and 1024 dimensions. While increasing $d$ can be seen to improve orthogonality, none are fully orthogonal. Note CIDER dot products appear more uniform than OPL, but the overall average absolute dot products do not differ much in Table 1.

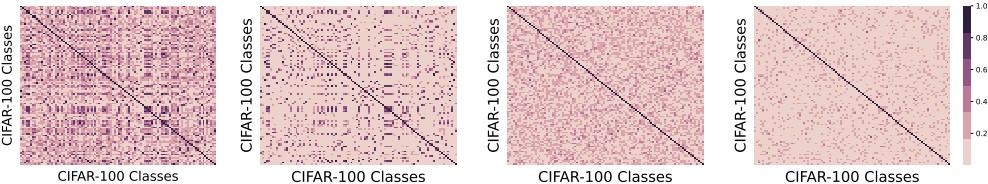

Figure 4: Orthogonality visualization of the dot product of the average per-class feature vectors. From Left to right: OPL(64), OPL(1024), CIDER(64), CIDER(1024).

Thus, OPL and CIDER cannot achieve complete orthogonality of different class features by clustering of the same class features. As one of our goals is to translate class indicators to align exactly onto coordinates for interpretability, these loss-function based approaches are not sufficient.

**Augmentation of OPL features with ICR and DCR**. Next we add our rectification algorithms, ICR and DCR, on top of the near-orthogonal, and compact embeddings as output by OPL or CIDER. We use $d = 1024$ as default, but also show the dimensions used in the paper as OPL(64) and CIDER(512). The orthogonality and compactness results are in Table 3. For ICR, we show the result after each of 5 iterations. Note that ICR improves the average dot product by about 1 digit of precision each iteration, and compactness stays about the same, sometimes increasing. DCR achieves two digits of precision in the average dot-product after one step, with a slight degradation in compactness.

## 3.2 CLASSIFICATION ACCURACY AFTER ICR/DCR

We next investigate the effect on classification accuracy after applying ICR and DCR. We now note that there are two standard ways in this setting to enact class predictions. The first is recommended in the OPL paper: build a simple logistic regression for each class, and choose the class with highest score for a query (denoted Smax). In this paper we prefer using the less powerful model of the Rocchio algorithm $\hat{j} = \arg\max_{j \in [k]} \langle v_j, q \rangle$, for a query $q$ (denoted NN). Table 4 shows the top-1 and top-5 accuracy for OPL, CIDER, and after applying +DCR or +ICR for up to 5 iterations.

Table 2: Softmax Top 1 and Top 5 Accuracy of each d

| Loss | | 64 dim | 128 dim | 256 dim | 512 dim | 1024 dim |
|------|-------|--------|---------|---------|---------|----------|
| OPL | Top 1 | 73.38 | 74.29 | 74.26 | 74.87 | 75.22 |
| CIDER | Top 1 | 71.94 | 72.23 | 72.00 | 72.00 | 71.80 |
| OPL | Top 5 | 91.41 | 92.42 | 92.61 | 92.62 | 92.14 |
| CIDER | Top 5 | 89.02 | 89.35 | 89.15 | 89.20 | 88.84 |

Table 3: Orthogonality and Compactness scores for OPL, CIDER, and each after applying +DCR or +ICR $j$, for $j$ iterations. As default with 1024 dimensions.

| Score | OPL (64) | OPL | OPL+DCR | OPL+ICR 1 | OPL+ICR 2 | OPL+ICR 3 | OPL+ICR 4 | OPL+ICR 5 |
|-------|----------|-----|---------|-----------|-----------|-----------|-----------|-----------|
| Orthogonality | 0.2709 | 0.1268 | 0.0015 | 0.0056 | 0.0006 | $8.2321e\text{-}5$ | $1.1560e\text{-}5$ | $1.7660e\text{-}6$ |
| Compactness | 0.9742 | 0.9899 | 0.9669 | 0.9785 | 0.9779 | 0.9779 | 0.9779 | 0.9779 |

| Score | CIDER (512) | CID | CID+DCR | CID+ICR 1 | CID+ICR 2 | CID+ICR 3 | CID+ICR 4 | CID+ICR 5 |
|-------|-------------|-----|---------|-----------|-----------|-----------|-----------|-----------|
| Orthogonality | 0.1017 | 0.0930 | 0.0057 | 0.0138 | 0.0021 | 0.0004 | $7.4106e\text{-}5$ | $1.5946e\text{-}5$ |
| Compactness | 0.9764 | 0.9754 | 0.9594 | 0.9586 | 0.9566 | 0.9563 | 0.9562 | 0.9562 |

For both the Smax (logistic) and NN (Rocchio) classifiers, the OPL initialization outperforms CIDER. Unsurprisingly, the more powerful Smax (logistic) classifier (about 75.2% on top-1) has a bit better performance than the NN (Rocchio) approach (about $74.5 - 75\%$ on top-1). The overall best score is found with just OPL ($d = 1024$) at 75.28% improving upon the baseline OPL ($d = 64$) at 73.20%; applying ICR slightly decreases this to 75.21% or 75.20%. However, for the NN classifier, applying ICR actually improves the score from OPL ($d = 64$) at 72.36% and OPL ($d = 1024$) at 74.57% up to a score of 75.03% – which is not far from the best Smax (logistic) score. Similar effects are seen with top-5 accuracy (and CIFAR-10 in Appendix B), where OPL outperforms CIDER, and in this case using ICR has little effect and provides improvement in NN (Rocchio) classifiers.

To verify that OPL+ISR does not deteriorate representations, we applied it to the training data (see Tables 13 and 14 in Appendix C) where all methods achieve between 99.5% and 100% accuracy; with the exception of some degradation under the Smax (logistic) classifier after using CIDER loss.

### 3.3 OUT-OF-DISTRIBUTION DETECTION

Out-of-Distribution Detection (OOD) is the task of identifying testing samples that originate from an unknown distribution, which data representation did not encountered during training. This task evaluates the model's dependability when encountering both known in-distribution (ID) inputs and OOD samples – these should not be forced into an existing classification structure, and may represent anomalies requiring further attention. A wide variety of OOD detection methods have been explored, with distance-based OOD detection demonstrating considerable potential Lee et al. (2018); Xing et al. (2019) via representation learning. A central approach extends a Rocchio-type set up and determines ID vs. OOD based on the distance to class means. Very recently Ming et al. (2023) introduced CIDER, a Compactness and Dispersion Regularized learning framework for OOD detection, discussed earlier in equation 2. This provides a significant improvement in the state of the art.

**Datasets and Training Details** In line with the approach taken by Ming et al. (2023), we adopt the CIFAR-10 and CIFAR-100 Krizhevsky (2009) as the in-distribution datasets (CIFAR-10 results in Appendix B). For evaluating the OOD detection performance, we use a diverse collection of natural image datasets encompassing SVHN Netzer et al. (2011), Places365 Zhou et al. (2018), Textures Cimpoi et al. (2013), LSUN Yu et al. (2015), and iSUN Xu et al. (2015); ($\star$) for space, we only show iSUN and Texture in Appendix D. In our experiments, we utilize the pre-trained ResNet-9 used in the Image Classification task for the CIFAR-100 dataset. We freeze the pre-trained model up to the penultimate layer to extract CIDER ID and OOD features for our OOD detection experiments. After obtaining the extracted CIDER features, we apply ICR to further refine the features, enhancing inter-class separation within the feature embedding space. Upon acquiring the ICR-rectified CIDER ID and OOD features at test time, we employ CIDER's distance-based code for OOD detection.

Table 4: Test data results for OPL, CIDER and + DCR or +ICR with 1024 dimensions

| Metric | OPL(64) | OPL | OPL+DCR | OPL+ICR 1 | OPL+ICR 2 | OPL+ICR 3 | OPL+ICR 4 | OPL+ICR 5 |
|---|---|---|---|---|---|---|---|---|
| Smax Top 1 | 73.20 | 75.28 | 74.47 | 75.21 | 75.19 | 75.19 | 75.20 | 75.20 |
| Smax Top 5 | 91.23 | 91.93 | 89.31 | 91.71 | 91.35 | 91.28 | 91.29 | 91.29 |
| NN Top 1 | 72.36 | 74.57 | 73.39 | 75.02 | 75.03 | 75.03 | 75.03 | 75.03 |
| NN Top 5 | 90.17 | 89.84 | 89.25 | 91.76 | 91.35 | 91.26 | 91.24 | 91.23 |
| Metric | CIDER (512) | CID | CID+DCR | CID+ICR 1 | CID+ICR 2 | CID+ICR 3 | CID+ICR 4 | CID+ICR 5 |
| Smax Top 1 | 72.00 | 71.80 | 71.46 | 71.59 | 71.60 | 71.58 | 71.58 | 71.79 |
| Smax Top 5 | 89.20 | 88.84 | 86.02 | 88.26 | 87.72 | 87.60 | 87.60 | 87.67 |
| NN Top 1 | 72.19 | 71.74 | 71.50 | 71.60 | 71.66 | 71.61 | 71.61 | 71.61 |
| NN Top 5 | 89.08 | 88.65 | 85.95 | 88.240 | 87.63 | 87.52 | 87.47 | 87.47 |

Table 5: OOD performance for CIDER, CIDER+DCR/ICR on CIFAR-100

| | SVHN | | Places365 | | LSUN | | Average ⋆ | |
|---|---|---|---|---|---|---|---|---|
| | FPR↓ | AUROC↑ | FPR↓ | AUROC↑ | FPR↓ | AUROC↑ | **FPR↓** | **AUROC↑** |
| CE+SimCLR | 24.82 | 94.45 | 86.63 | 71.48 | 56.40 | 89.00 | 59.62 | 84.15 |
| KNN+ | 39.23 | 92.78 | 80.74 | 77.58 | 48.99 | 89.3 | 60.22 | 86.14 |
| OPL | 98.83 | 43.00 | 99.16 | 38.08 | 99.85 | 25.93 | 96.18 | 44.42 |
| CIDER | 44.16 | 89.47 | 69.44 | 80.82 | 57.59 | 86.29 | 43.24 | 89.28 |
| CIDER+DCR | 48.52 | 88.21 | 71.29 | 79.95 | 62.18 | 84.33 | 46.05 | 88.25 |
| CIDER+ICR 1 | 49.28 | 87.97 | 70.28 | 79.93 | 60.42 | 84.94 | 45.75 | 88.32 |
| CIDER+ICR 2 | 49.72 | 87.92 | 70.53 | 79.89 | 60.51 | 84.86 | 45.97 | 88.27 |

The following metrics are reported in Table 5: 1) False Positive Rate (FPR) of OOD samples at 95% True Positive Rate (TPR) of ID samples, and 2) Area Under the Receiver Operating Characteristic Curve (AUROC). We show two representative prior art: CE+SimCLR Winkens et al. (2020) and KNN+Sun et al. (2022), the best two methods before CIDER. Observe how CIDER significantly improves FPR from about 60% to about 43% and AUROC from about 84-86% to 89% (averaged across data sets). Applying ICR or DCR shows a small degradation of these improvements, with a FPR about 45% and AUROC about 88%, still a significant improvement over the previous baselines, but now with interpretable structure. On CIFAR-10 CIDER+ICR slightly improves over just CIDER; see Appendix B. This tasks seems delicate, and for instance using OPL equation 1 in place of CIDER equation 2 achieves much worse results with an average FPR of 96% and AUROC of only 44%.

## 4    CONCLUSION & DISCUSSION

This paper introduces a post-processing to the training phase of a learned embedding mechanism which provides interpretable structure. Namely, for a learned embedded representation for a multi-class classification task, our method Iterative Class Rectification (ICR) continuously adjusts the embedding function so each of $k$ identified class means is associated with a coordinate. Thus the representation of each class is orthogonal, and can be independently measured. This does not preclude an object from having an association with multiple classes, but it decouples those contributions.

This class orthogonality could also be useful if the class is associated with a protected class (e.g., gender, race, etc). By restricting to classifiers which predict labels based on dot products along these class coordinates, we could eliminate association learned about that trait by simply ignoring that coordinate from the representation at the evaluation phase. This pre-processes and makes simple the technique that has become popular in language debiasing Bolukbasi et al. (2016); Dev & Phillips (2019); Ravfogel et al. (2020); Wang et al. (2020) which first attempts to identify a linear subspace, and then projects all data in the representation off that subspace.

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

# A    PROOFS OF CONVERGENCE OF BINARYICR

## A.1    GRADED ROTATION

For an angle $\theta$ we denote the $2 \times 2$ rotation matrix by $R_\theta$, that is, $R_\theta = \begin{bmatrix} \cos\theta & -\sin\theta \\ \sin\theta & \cos\theta \end{bmatrix}$. The graded rotation of a vector $x$ with respect to the mean vectors $v_1$ and $v_2$ was introduced in Dev et al. (2021); Aboagye et al. (2023), which we recall below.

---

**Algorithm 5** GradedRotat($v_1, v_2, x$)

---

1: **Input:** Unit vectors $v_1$, $v_2$ in $\mathbb{R}^2$ and $x \in \mathbb{R}^2$
2: Set $\theta' = \arccos(\langle v_1, v_2 \rangle)$ and $\theta = \frac{\pi}{2} - \theta'$
3: Set $\phi_1 = \arccos \langle v_1, \frac{x}{|x|} \rangle$
4: Set $v_2' = v_2 - \langle v_1, v_2 \rangle v_1$
5: Set $d_2 = \arccos \langle v_2', \frac{x}{|x|} \rangle$
6: Compute $\theta_x = \begin{cases} \theta \frac{\phi_1}{\theta'} & \text{if } d_2 > 0 \text{ and } \phi_1 \leq \theta' \\ \theta \frac{\pi - \phi_1}{\pi - \theta'} & \text{if } d_2 > 0 \text{ and } \phi_1 > \theta' \\ -\theta \frac{\pi - \phi_1}{\theta'} & \text{if } d_2 < 0 \text{ and } \phi_1 \geq \pi - \theta' \\ -\theta \frac{\phi_1}{\pi - \theta'} & \text{if } d_2 < 0 \text{ and } \phi_1 < \pi - \theta' \end{cases}$
7: **return** $R_{\theta_x} x$

---

## A.2    CONVERGENCE OF BINARYICR AND BINARYDCR

In order to prove the convergence of BinaryICR and BinaryDCR algorithms, we need to make the following assumptions on data, which are illustrated in Figure 5. Notice Assumption 1 is a special case of Assumption 2.

**Assumption 2** *Let* $0 < \theta' < \frac{\pi}{2}$, $\theta = \frac{\pi}{2} - \theta'$, $-\phi_1 \leq 0 \leq \phi_2 < \psi_1 \leq \psi_2 \leq \pi$ *and* $\gamma = \psi_1 - \phi_2 > 0$. *Let also* $X_1$ *and* $X_2$ *be subsets of the cones* $\Gamma_1 = \{re^{i\phi} : \phi_1 \leq \phi \leq \phi_2, r > 0\}$ *and* $\Gamma_2 = \{re^{i\psi} : \psi_1 \leq \psi \leq \psi_2, r > 0\}$, *respectively, and* $\theta'$ *be the angle between the mean vectors* $v_1$ *and* $v_2$ *of* $X_1$ *and* $X_2$, *respectively (see Figure 5).*

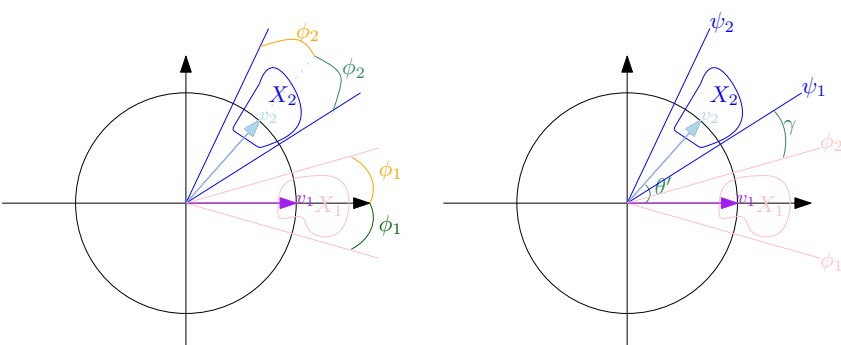

Figure 5: Pictorial view of Assumption 1 (left) and Assumption 2 (right) on two classes $X_1$ and $X_2$.

**Lemma 4** *Under Assumption 2, for any* $i$, *the angle* $\theta_i = \frac{\pi}{2} - \theta_i'$ *stays positive, where* $\theta_1' = \theta'$ *and* $\theta_i'$ $(i \geq 2)$ *shows the angle between two class means after* $i$-*th iteration of BinaryICR.*

*Proof.* First we discuss what happens for the cones $\Gamma_1$ and $\Gamma_2$ after one iteration of BinaryICR. Then by an induction argument we conclude that $\theta_i > 0$ for any $i$.

In $\Gamma_1$, the half-cone under $v_1$ shrinks but the other half expands. It means that the $y$-values of data points in the half-cone under $v_1$ increases and their $x$-values decreases a bit but stays positive. The

same phenomenon happens for the other half of the cone. Since we had increase in $y$-values, their average will increase as well (i.e. will be positive as it was 0 previously). Therefore, $v_1'$ will be in the first quadrant.

For $\Gamma_2$, after running one iteration of BinaryICR the half-cone above $v_2$ shrinks but the other half expands. Now in order to make comparison easy, we rotate all the points of $X_2'$ by $-\theta_1$ (i.e. $y$-axis will be transformed on top of $v_2$) and call it $X_2''$, where $X_2'$ is the transformation of $X_2$ after applying graded rotations on top of $X_2$. This means that the $x$-values of data points in $X_2''$ are increased with respect to their $x$-values in $X_2$ (consider two cases $\psi_2 \leq \pi/2$ and $\psi_2 > \pi/2$ separately). This will be happened to their average as well, and thus their average will be under $v_2$. Rotating the points of $X_2''$ back by $\theta_1$ degree to get $X_2'$, it means that the average of $X_2'$, which we call $v_2'$, will be less than $\pi/2$.

We observe that both mean vectors $v_1'$ and $v_2'$ stay in the first quadrant, implying $\theta_2' < \pi/2$ or equivalently $\theta_2 > 0$.

Now by an induction argument if $\theta_i > 0$, completely similar to going from $\theta_1 > 0$ to $\theta_2 > 0$ above, we can conclude that $\theta_{i+1} > 0$. □

**Theorem 5 (Restatement of Theorem 1)** *Under Assumption 2 the BinaryICR algorithm converges, that is, after enough epochs, $\theta_i'$ will approach to $\frac{\pi}{2}$, where $\theta_i'$ is the angle between two class means after $i$th iteration of BinaryICR.*

*Proof.* Let $\theta_1 = \theta$ and $\theta_1' = \theta'$. Notice that all vectors in $(X_1 \cup X_2) \setminus \mathbb{R} \times \{0\}$ will be changed after any iteration of BinaryICR if $\theta_1' \neq \frac{\pi}{2}$. Now consider the gap $\gamma_1 = \gamma$ between $\phi_2$ and $\psi_1$, i.e. $\gamma_1 = \psi_1 - \phi_2 > 0$. Since both $\phi_2$ and $\psi_1$ lie in $[0, \theta_1']$, after one iteration of BinaryICR, they will be mapped to $\phi_2 + \frac{\theta_1}{\theta_1'}\phi_2$ and $\psi_1 + \frac{\theta_1}{\theta_1'}\psi_1$. Thus $\gamma_1$ will be changed to $\gamma_2 = \gamma_1 + \frac{\theta_1}{\theta_1'}\gamma_1 > \gamma_1$ (note $\theta_i \geq 0$ and $0 < \gamma < \theta_i' < \pi/2$ by Lemma 4). Considering $\theta_2'$, running another iteration of BinaryICR will modify $\gamma_2$ to $\gamma_3 = \gamma_2 + \frac{\theta_2}{\theta_2'}\gamma_2 > \gamma_2$ and so on. Therefore, the sequence $(\gamma_n) \subset [0, \frac{\pi}{2}]$ is a bounded increasing sequence and thus convergent, say to $\gamma'$. This means that running another iteration of BinaryICR will not change $\gamma'$, that is $\theta_n \to 0$, otherwise $\gamma'$ will need to be changed. Hence, BinaryICR algorithm converges. □

**Theorem 6 (Restatement of Theorem 2)** *Assume Assumption 1 holds. In addition, if the angle between $v_1$ and $v_2$ is bigger than $\frac{\pi}{2}$, then we assume $\phi_1, \phi_2$ are less than $\frac{\pi}{4}$. Then after running the BinaryDCR algorithm, the class means will be orthogonal to each other.*

*Proof.* The proof is trivial, but we include it for completeness. Let $\theta'$ be the angle between $v_1$ and $v_2$. There are two cases.

*Case 1.* When $\theta' < \frac{\pi}{2}$ and the two classes are disjoint, according to the BinaryDCR algorithm, all vectors in class 2 will be rotated by $\theta = \frac{\pi}{2} - \theta'$ degrees, and so their mean $v_2$ will be rotated by $\theta$ degrees as well. However, the vectors in class 1 will not be rotated. Thus $v_1$ will stay the same. Therefore, after running the algorithm, the class means will be orthogonal to each other.

*Case 2.* In the case $\theta' > \frac{\pi}{2}$, according to the BinaryDCR algorithm, all the vectors within $\frac{\pi}{4}$ of $v_2$ will be rotated by $\theta = \frac{\pi}{4}$ degrees. So, by the assumptions all the points in class 2, and thus their mean $v_2$, will be rotated by $\frac{\pi}{4}$ degrees. Since the points in class 1 will stay the same, this means that, after running the algorithm, the class means will be orthogonal to each other. □

## A.3 CONVERGENCE OF MULTIDCR

**Assumption 3** *We consider the following assumptions on the dataset in order to prove convergence of MultiClassDCR algorithm (see Figure 6). Without loss of generality we assume if we run the Gram-Schmidt process on class means $\{v_1, \ldots, v_k\}$, and it runs successfully (handled by assumption (1)), then the resulting orthonormal basis would be the standard basis $\{e_1, \ldots, e_k\}$.*

1. *Class means are linearly independent.*

2. *For each $i$, class $X_i$ is included in a cone $C_i$ around $v_i$ with radius $\phi_i$, where for $i \geq 3$, $C_i$ is located inside a cone $B_i$ around $e_i$ of radius less than $\pi/2$.*

3. *All class means are in the first orthant, or $\phi_i < \frac{\pi}{4}$ for all $i$.*

4. *For all $j < i$, where $i \geq 3$, $X_j$ is outside of the cone $B_i$.*

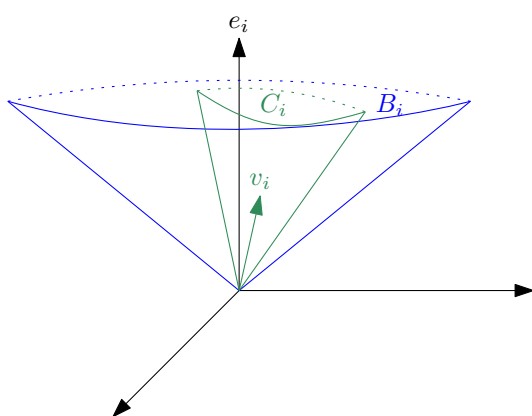

Figure 6: Pictorial view of Assumption 3.

**Theorem 7 (Restatement of Theorem 3)** *Let Assumption 3 hold. Then after running the MultiDCR algorithm all class means will be orthogonal to each other.*

*Proof.* In MultiDCR algorithm for each class we rotate the encompassing cone in a Gram-Schmidt manner. Considering the separation assumptions and linear independence property in Assumption 3, after any step in Gram-Schmidt process, all cones will stay separated. This is because in Gram-Schmidt process, we orthogonalize vectors one by one; notice that this process happens in the same subspace as before, that is, in the $i$th step the span of $\{e_1, \ldots, e_i\}$ and $\{v_1, \ldots, v_i\}$ will be the same. Now Assumption 3 implies that the $i$th class cone $C_i$ around $v_i$ will be rotated in such a way so that $e_i$ will be its center after the rotation. We call this rotated cone $C_i'$. Thus $C_i'$ will be inside the cone $B_i$, by Assumption 3(2). This means that the $C_i'$ will be disjoint from the previously orthogonalized cones $C_j'$ for $j < i$ as they live outside the cone $B_i$ and so outside the cone $C_i'$. Therefore, after running the MultiDCR algorithm, all class means will be orthogonal to each other. $\square$

## B EXPERIMENTS ON CIFAR-10

### B.1 ORTHOGONALITY AND COMPACTNESS

Table 6: Average class dot products; and intra-class compactness.

| dim: | 64 | 128 | 256 | 512 | 1024 | 64 | 128 | 256 | 512 | 1024 |
|------|------|------|------|------|------|------|------|------|------|------|
| OPL | 0.0111 | 0.0093 | 0.0083 | 0.0036 | 0.0058 | 0.9989 | 0.9990 | 0.9990 | 0.9990 | 0.9991 |
| CIDER | 0.1111 | 0.1111 | 0.1111 | 0.1111 | 0.1111 | 0.9892 | 0.9885 | 0.9880 | 0.9875 | 0.9859 |

Table 7: Softmax Top 1 and Top 5 Accuracy of each k

| Loss | | 64 dim | 128 dim | 256 dim | 512 dim | 1024 dim |
|------|------|--------|---------|---------|---------|----------|
| OPL | Top 1 | 93.020 | 93.610 | 93.200 | 93.420 | 93.310 |
| CIDER | Top 1 | 92.730 | 92.640 | 92.870 | 92.730 | 92.590 |
| OPL | Top 5 | 99.590 | 99.570 | 99.610 | 99.570 | 99.650 |
| CIDER | Top 5 | 99.570 | 99.590 | 99.550 | 99.520 | 99.690 |

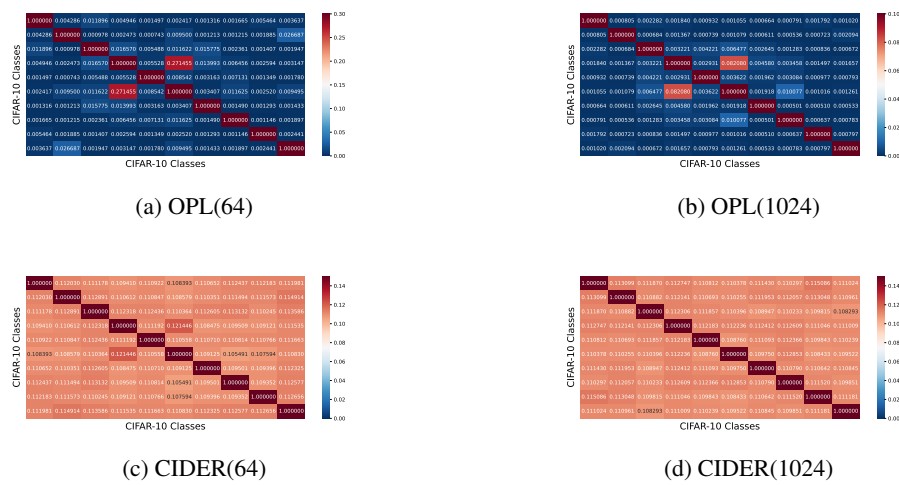

Figure 7: Orthogonality visualization of the dot product of the average per-class feature vectors. From Left to right: OPL(64), OPL(1024), CIDER(64), CIDER(1024).

## B.2 ORTHOGONALITY VISUALIZATION

## B.3 AUGMENTATION OF OPL FEATURES WITH ICR AND DCR

Table 8: Orthogonality and Compactness scores for OPL, CIDER, and each after applying +DCR or +OPL $j$, for $j$ iterations. As default with 1024 dimensions.

| Score | OPL (64) | OPL | OPL+DCR | OPL+ICR 1 | OPL+ICR 2 | OPL+ICR 3 | OPL+ICR 4 | OPL+ICR 5 |
|---|---|---|---|---|---|---|---|---|
| Orthogonality | 0.0111 | 0.0036 | $2.0720e\text{-}05$ | $5.2844e\text{-}05$ | $1.0261e\text{-}06$ | $1.8735e\text{-}08$ | $3.2816e\text{-}10$ | $5.7593e\text{-}12$ |
| Compactness | 0.9989 | 0.9991 | 0.9991 | 0.9991 | 0.9991 | 0.9991 | 0.9991 | 0.9991 |

| Score | CIDER (512) | CID | CID+DCR | CID+ICR 1 | CID+ICR 2 | CID+ICR 3 | CID+ICR 4 | CID+ICR 5 |
|---|---|---|---|---|---|---|---|---|
| Orthogonality | 0.1111 | 0.1111 | 0.0744 | 0.0838 | 0.0592 | 0.0372 | 0.0239 | 0.0143 |
| Compactness | 0.9875 | 0.9859 | 0.9779 | 0.9351 | 0.8976 | 0.8778 | 0.8662 | 0.8601 |

## B.4 CLASSIFICATION ACCURACY AFTER ICR/DCR ON TEST DATA

Table 9: Test data results for OPL, CIDER and + DCR or +ICR with 1024 dimensions

| Metric | OPL(64) | OPL | OPL+DCR | OPL+ICR 1 | OPL+ICR 2 | OPL+ICR 3 | OPL+ICR 4 | OPL+ICR 5 |
|---|---|---|---|---|---|---|---|---|
| Smax Top 1 | 93.020 | 93.310 | 93.330 | 93.330 | 93.330 | 93.330 | 93.330 | 93.330 |
| Smax Top 5 | 99.590 | 99.650 | 98.700 | 98.700 | 98.700 | 98.700 | 98.700 | 98.700 |
| NN Top 1 | 93.030 | 93.300 | 93.290 | 93.300 | 93.300 | 93.300 | 93.300 | 93.300 |
| NN Top 5 | 99.560 | 99.720 | 98.900 | 98.920 | 98.920 | 98.920 | 98.920 | 98.920 |

| Metric | CIDER (512) | CID | CID+DCR | CID+ICR 1 | CID+ICR 2 | CID+ICR 3 | CID+ICR 4 | CID+ICR 5 |
|---|---|---|---|---|---|---|---|---|
| Smax Top 1 | 92.730 | 92.590 | 92.360 | 92.630 | 92.610 | 92.440 | 92.370 | 92.400 |
| Smax Top 5 | 99.520 | 99.690 | 99.180 | 99.630 | 99.610 | 99.580 | 99.570 | 99.590 |
| NN Top 1 | 92.730 | 92.560 | 92.180 | 92.420 | 91.210 | 90.000 | 89.940 | 89.940 |
| NN Top 5 | 99.420 | 99.550 | 99.010 | 99.170 | 98.870 | 96.840 | 95.510 | 95.350 |

## B.5 Classification Accuracy after ICR/DCR on Training Data

Table 10: Training data results for OPL, OPL+DCR, and OPL+ICR with 1024 dimensions

| Metric | OPL (64) | OPL (1024) | OPL+DCR | OPL+ICR1 | OPL+ICR2 | OPL+ICR3 | OPL+ICR4 | OPL+ICR5 |
|---|---|---|---|---|---|---|---|---|
| Smax Top 1 | 99.976 | 99.976 | 99.976 | 99.976 | 99.976 | 99.976 | 99.976 | 99.976 |
| Smax Top 5 | 99.998 | 100.000 | 100.000 | 100.000 | 100.000 | 100.000 | 100.000 | 100.000 |
| NN Top 1 | 99.974 | 93.300 | 93.290 | 99.974 | 99.974 | 99.974 | 99.974 | 99.974 |
| NN Top 5 | 100.000 | 99.720 | 98.900 | 100.000 | 100.000 | 100.000 | 100.000 | 100.000 |

Table 11: Training data results for CIDER, CIDER+DCR, and CIDER+ICR with 1024 dimensions

| Metric | CIDER (512) | CID | CID+DCR | CID+ICR 1 | CID+ICR 2 | CID+ICR 3 | CID+ICR 4 | CID+ICR 5 |
|---|---|---|---|---|---|---|---|---|
| Smax Top 1 | 99.944 | 99.952 | 99.150 | 94.176 | 93.402 | 87.858 | 86.980 | 86.890 |
| Smax Top 5 | 100.000 | 100.000 | 99.998 | 98.456 | 93.588 | 93.582 | 93.192 | 87.336 |
| NN Top 1 | 99.946 | 99.950 | 99.716 | 99.872 | 98.178 | 96.554 | 96.478 | 96.474 |
| NN Top 5 | 100.000 | 100.000 | 100.000 | 100.000 | 99.926 | 97.570 | 96.590 | 96.572 |

## B.6 FULL TABLE FOR OUT OF DISTRIBUTION EXPERIMENT USING CIFAR-10 AS IN-DISTRIBUTION (ID) DATA

Table 12: OOD performance for for CIDER, CIDER+DCR, and CIDER+ICR on the CIFAR10 Dataset

| Method | SVHN | | Places365 | | LSUN | | iSUN | | Texture | | Average | |
|---|---|---|---|---|---|---|---|---|---|---|---|---|
| | FPR↓ | AUROC↑ | FPR↓ | AUROC↑ | FPR↓ | AUROC↑ | FPR↓ | AUROC↑ | FPR↓ | AUROC↑ | FPR↓ | AUROC↑ |
| **Without Contrastive Learning** | | | | | | | | | | | | |
| MSP | 59.66 | 91.25 | 62.46 | 88.64 | 45.21 | 93.80 | 54.57 | 92.12 | 66.45 | 88.50 | 57.67 | 90.86 |
| ODIN | 53.78 | 91.30 | 43.40 | 90.98 | 10.93 | 97.93 | 28.44 | 95.51 | 55.59 | 89.47 | 38.43 | 93.04 |
| Mahalanobis | 9.24 | 97.80 | 83.50 | 69.56 | 67.73 | 73.61 | 6.02 | 98.63 | 23.21 | 92.91 | 37.94 | 86.50 |
| Energy | 54.41 | 91.22 | 42.77 | 91.02 | 10.19 | 98.05 | 27.52 | 95.59 | 55.23 | 89.37 | 38.02 | 93.05 |
| GODIN | 18.72 | 96.10 | 55.25 | 85.50 | 11.52 | 97.12 | 30.02 | 94.02 | 33.58 | 92.20 | 29.82 | 92.97 |
| LogitNorm | | | | | | | | | | | | |
| **With Contrastive Learning** | | | | | | | | | | | | |
| ProxyAnchor | 39.27 | 94.55 | 43.46 | 92.06 | 21.04 | 97.02 | 23.53 | 96.56 | 42.70 | 93.16 | 34.00 | 94.67 |
| CE+SimCLR | 6.98 | 99.22 | 54.39 | 86.70 | 64.53 | 85.60 | 59.62 | 86.78 | 16.77 | 96.56 | 40.46 | 90.97 |
| CSI | 37.38 | 94.69 | 38.31 | 93.04 | 10.63 | 97.93 | 10.36 | 98.01 | 28.85 | 94.87 | 25.11 | 95.71 |
| SSD+ | 2.47 | 99.51 | 22.05 | 95.57 | 10.56 | 97.83 | 28.44 | 95.67 | 9.27 | 98.35 | 14.56 | 97.38 |
| KNN+ | 2.70 | 99.61 | 23.05 | 94.88 | 7.89 | 98.01 | 24.56 | 96.21 | 10.11 | 97.43 | 13.66 | 97.22 |
| **Regularization for Compactness and Dispersion** | | | | | | | | | | | | |
| CIDER | 8.30 | 98.46 | 21.37 | 95.93 | 9.63 | 98.18 | 0.68 | 99.79 | 27.75 | 94.45 | 13.55 | 97.36 |
| CIDER+DCR | 8.33 | 98.46 | 21.58 | 95.92 | 9.59 | 98.19 | 0.67 | 99.79 | 27.75 | 94.48 | 13.58 | 97.37 |
| CIDER+ICR 1 | 8.31 | 98.46 | 21.33 | 95.93 | 9.48 | 98.19 | 0.69 | 99.79 | 27.82 | 94.44 | 13.53 | 97.36 |
| CIDER+ICR 2 | 8.32 | 98.46 | 21.29 | 95.93 | 9.46 | 98.19 | 0.69 | 99.79 | 27.84 | 94.45 | 13.52 | 97.36 |
| CIDER+ICR 3 | 8.32 | 98.46 | 21.30 | 95.93 | 9.46 | 98.19 | 0.69 | 99.79 | 27.84 | 94.45 | 13.52 | 97.36 |
| CIDER+ICR 4 | 8.32 | 98.46 | 21.29 | 95.93 | 9.46 | 98.19 | 0.69 | 99.79 | 27.84 | 94.45 | 13.52 | 97.36 |
| CIDER+ICR 5 | 8.32 | 98.46 | 21.29 | 95.93 | 9.46 | 98.19 | 0.69 | 99.79 | 27.84 | 94.45 | 13.52 | 97.36 |
| OPL | 99.74 | 33.74 | 99.44 | 42.56 | 99.75 | 56.45 | 99.93 | 49.33 | 97.15 | 40.49 | 99.20 | 44.51 |

*OOD Dataset*

## C  MORE EXPERIMENTS ON ACCURACY FOR CIFAR-100

Table 13: Training data results for OPL, OPL+DCR, and OPL+ICR with 1024 dimensions

| Metric | OPL (64) | OPL (1024) | OPL+DCR | OPL+ICR1 | OPL+ICR2 | OPL+ICR3 | OPL+ICR4 | OPL+ICR5 |
|---|---|---|---|---|---|---|---|---|
| Smax Top 1 | 99.946 | 99.976 | 99.762 | 99.594 | 99.686 | 99.698 | 99.698 | 99.698 |
| Smax Top 5 | 100.000 | 100.000 | 99.992 | 100.000 | 100.000 | 100.000 | 100.000 | 100.000 |
| NN Top 1 | 99.858 | 99.972 | 99.222 | 99.974 | 99.974 | 99.974 | 99.974 | 99.974 |
| NN Top 5 | 100.000 | 100.000 | 100.000 | 100.000 | 100.000 | 100.000 | 100.000 | 100.000 |

Table 14: Training data results for CIDER, CIDER+DCR, and CIDER+ICR with 1024 dimensions

| Metric | CIDER (512) | CID | CID+DCR | CID+ICR 1 | CID+ICR 2 | CID+ICR 3 | CID+ICR 4 | CID+ICR 5 |
|---|---|---|---|---|---|---|---|---|
| Smax Top 1 | 99.888 | 99.892 | 94.370 | 97.396 | 84.540 | 82.798 | 82.612 | 82.572 |
| Smax Top 5 | 100.000 | 100.000 | 98.266 | 99.408 | 89.178 | 88.088 | 87.920 | 87.902 |
| NN Top 1 | 99.890 | 99.898 | 99.720 | 99.872 | 99.864 | 99.862 | 99.862 | 99.862 |
| NN Top 5 | 100.000 | 100.000 | 99.988 | 100.000 | 99.998 | 99.998 | 99.998 | 99.998 |

# D FULL TABLE FOR OUT OF DISTRIBUTION EXPERIMENT USING CIFAR-100 AS IN-DISTRIBUTION (ID) DATA

Table 15: OOD performance for for CIDER, CIDER+DCR, and CIDER+ICR on the CIFAR100 Dataset

| Method | SVHN FPR↓ | SVHN AUROC↑ | Places365 FPR↓ | Places365 AUROC↑ | LSUN FPR↓ | LSUN AUROC↑ | iSUN FPR↓ | iSUN AUROC↑ | Texture FPR↓ | Texture AUROC↑ | Average FPR↓ | Average AUROC↑ |
|---|---|---|---|---|---|---|---|---|---|---|---|---|
| **Without Contrastive Learning** | | | | | | | | | | | | |
| MSP | 78.89 | 79.8 | 84.38 | 74.21 | 83.47 | 75.28 | 84.61 | 74.51 | 86.51 | 72.53 | 83.12 | 75.27 |
| ODIN | 70.16 | 84.88 | 82.16 | 75.19 | 76.36 | 80.1 | 79.54 | 79.16 | 85.28 | 75.23 | 78.7 | 79.11 |
| Mahalanobis | 87.09 | 80.62 | 84.63 | 73.89 | 84.15 | 79.43 | 83.18 | 78.83 | 61.72 | 84.87 | 80.15 | 79.53 |
| Energy | 66.91 | 85.25 | 81.41 | 76.37 | 59.77 | 86.69 | 66.52 | 84.49 | 79.01 | 79.96 | 70.72 | 82.55 |
| GODIN | 74.64 | 84.03 | 89.13 | 68.96 | 93.33 | 67.22 | 94.25 | 65.26 | 86.52 | 69.39 | 87.57 | 70.97 |
| LogitNorm | 59.6 | 90.74 | 80.25 | 78.58 | 81.07 | 82.99 | 84.19 | 80.77 | 86.64 | 75.6 | 78.35 | 81.74 |
| **With Contrastive Learning** | | | | | | | | | | | | |
| ProxyAnchor | 87.21 | 82.43 | 70.1 | 79.84 | 37.19 | 91.68 | 70.01 | 84.96 | 65.64 | 84.99 | 66.03 | 84.78 |
| CE+SimCLR | 24.82 | 94.45 | 86.63 | 71.48 | 56.4 | 89 | 66.52 | 83.82 | 63.74 | 82.01 | 59.62 | 84.15 |
| CSI | 44.53 | 92.65 | 79.08 | 76.27 | 75.58 | 83.78 | 76.62 | 84.98 | 61.61 | 86.47 | 67.48 | 84.83 |
| SSD+ | 31.19 | 94.19 | 77.74 | 79.9 | 79.39 | 85.18 | 80.85 | 84.08 | 66.63 | 86.18 | 67.16 | 85.9 |
| KNN+ | 39.23 | 92.78 | 80.74 | 77.58 | 48.99 | 89.3 | 74.99 | 82.69 | 57.15 | 88.35 | 60.22 | 86.14 |
| **Regularization for Compactness and Dispersion** | | | | | | | | | | | | |
| CIDER | 44.16 | 89.47 | 69.44 | 80.82 | 57.59 | 86.29 | 9.27 | 98.09 | 35.74 | 91.72 | 43.24 | 89.28 |
| CIDER+DCR | 48.52 | 88.21 | 71.29 | 79.95 | 62.18 | 84.33 | 10.78 | 97.8 | 37.46 | 90.95 | 46.05 | 88.25 |
| CIDER+ICR 1 | 49.28 | 87.97 | 70.28 | 79.93 | 60.42 | 84.94 | 10.96 | 97.71 | 37.84 | 91.02 | 45.75 | 88.32 |
| CIDER+ICR 2 | 49.72 | 87.92 | 70.53 | 79.89 | 60.51 | 84.86 | 11.08 | 97.7 | 38.03 | 90.99 | 45.97 | 88.27 |
| CIDER+ICR 3 | 49.82 | 87.9 | 70.6 | 79.88 | 60.59 | 84.84 | 11.15 | 97.7 | 38.07 | 90.98 | 46.05 | 88.26 |
| CIDER+ICR 4 | 49.85 | 87.9 | 70.61 | 79.87 | 60.62 | 84.84 | 11.15 | 97.7 | 38.16 | 90.98 | 46.08 | 88.26 |
| CIDER+ICR 5 | 49.84 | 87.9 | 70.57 | 79.87 | 60.59 | 84.84 | 11.15 | 97.7 | 38.12 | 90.98 | 46.05 | 88.26 |
| OPL | 98.83 | 43 | 99.16 | 38.08 | 99.85 | 25.93 | 91.52 | 63.2 | 91.54 | 51.9 | 96.18 | 44.42 |

# E  QUALITATIVE EXAMPLE COMPARISON

In Figure 8 we show a few illustrative examples from CIFAR-100, and compare their predictions on OPL, CIDER, and applying +DCR or +ICR. For each image we show that top 5 results under the NN (Rocchio) classifier. After ICR or DCR, these are the coordinates in the new coordinate system associated with the $k = 100$ classes.

We observe that while all methods have the label correct as the top prediction, the drop-off after the first prediction is steeper after ICR. For instance, because under OPL the class means for orange are correlated with other fruit (e.g., apple, pear), under OPL the orange example also has a high dot-product with those fruits. But after ICR or DCR, the class means are orthogonal and so the forced high association is gone. The same phenomenon can be see with man & boy and with woman & girl.

**WORM**

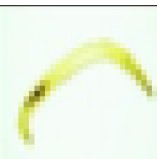

**OPL(64):** worm(98.81), snake(89.8), lizard(63.97), caterpillar(61.23), crab(59.34)
**OPL(1024):** worm(99.6), snake(73.7), plate(37.51), caterpillar(34.95), lizard(31.65)
**OPL + DCR:** worm(99.58), pear(2.57), sweet_pepper(1.55), orange(0.76), shrew(0.74)
**OPL + ICR:** worm(99.38), sweet_pepper(3.72), pear(3.32), cockroach(2.11), orange(1.81)

**CIDER(512):** worm(73.21), caterpillar(72.69), lizard(31.32), snake(29.45), rocket(25.37)
**CIDER(1024):** worm(88.74), caterpillar(43.52), ray(27.07), snake(23.22), road(19.6)
**CIDER + DCR:** worm(68.16), caterpillar(41.54), road(28.5), ray(25.51), sunflower(18.38)
**CIDER + ICR:** worm(77.79), caterpillar(37.37), ray(21.99), road(19.35), snail(11.55)

**ORANGE**

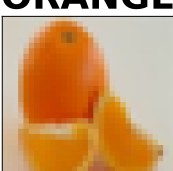

**OPL(64):** orange(99.75), apple(82.04), pear(78.8), sweet_pepper(72.15), poppy(45.62)
**OPL(1024):** orange(99.8), apple(80.54), pear(77.65), sweet_pepper(77.58), tulip(59.06)
**OPL + DCR:** orange(99.8), cup(0.65), bee(0.46), bicycle(0.42), cloud(0.37)
**OPL + ICR:** orange(99.65), cockroach(0.87), maple_tree(0.61), girl(0.59), orchid(0.51)

**CIDER(512):** orange(99.43), chair(20.74), crab(20.26), pear(19.86), sweet_pepper(19.18)
**CIDER(1024):** orange(99.83), pear(20.0), chair(19.31), poppy(17.53), bus(15.24)
**CIDER + DCR:** orange(99.69), house(2.2), butterfly(1.9), bear(1.75), skyscraper(1.3)
**CIDER + ICR:** orange(99.32), motorcycle(1.82), train(1.42), rose(1.34), forest(1.24)

**WOMAN**

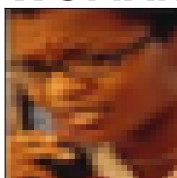

**OPL(64):** woman(96.37), girl(93.05), man(83.07), boy(81.57), baby(74.65)
**OPL(1024):** woman(98.67), girl(88.85), boy(88.26), man(87.61), baby(82.9)
**OPL + DCR:** woman(98.13), lobster(10.45), crab(5.05), dolphin(5.02), lamp(2.43)
**OPL + ICR:** woman(97.67), lobster(7.79), crab(4.62), wolf(2.45), lamp(2.01)

**CIDER(512):** woman(98.0), man(41.46), lamp(26.47), girl(25.69), camel(25.63)
**CIDER(1024):** woman(97.71), man(45.12), camel(35.11), pear(24.46), girl(21.77)
**CIDER + DCR:** woman(96.94), lamp(7.56), shark(7.05), couch(6.89), bee(5.48)
**CIDER + ICR:** woman(93.42), man(23.59), pear(9.9), palm_tree(6.85), lamp(6.51)

**MAN**

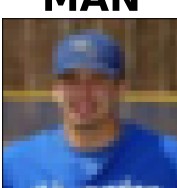

**OPL(64):** man(98.73), boy(89.51), woman(84.38), girl(78.22), baby(74.1)
**OPL(1024):** man(99.4), boy(92.82), girl(89.67), woman(89.63), baby(88.4)
**OPL + DCR:** man(99.32), boy(2.94), baby(2.41), streetcar(2.39), beaver(1.4)
**OPL + ICR:** man(97.75), boy(16.47), baby(6.99), bicycle(2.75), dolphin(1.9)

**CIDER(512):** man(99.06), boy(39.74), woman(35.32), flatfish(33.74), cattle(21.79)
**CIDER(1024):** man(96.26), boy(41.29), flatfish(35.21), lamp(29.47), palm_tree(18.43)
**CIDER + DCR:** man(95.46), turtle(12.15), lamp(11.19), flatfish(8.64), boy(7.17)
**CIDER + ICR:** man(96.42), lamp(7.87), porcupine(5.53), possum(5.36), bee(5.11)

Figure 8: Example images and top-5 scoring NN (Rocchio) values among classes in CIFAR-100.

