# OpenReview forum: "One-Hot Encoding Strikes Back: Fully Orthogonal Coordinate-Aligned Class Representations"
_ICLR.cc/2024/Conference — Submitted to ICLR 2024_

### Official Review · Reviewer_vbSf · 2023-10-14

**Soundness:** 2 fair
**Presentation:** 2 fair
**Contribution:** 2 fair
**Rating:** 3
**Confidence:** 4

**Summary:**

This paper proposes two techniques ICR and DCR for transforming existing class embeddings to be orthogonal and axis aligned for interpretability and better performance.

The brevity of the review doesn't stand for the quality of the review or of the paper, it is solely because of the prime questions I have about the paper.

**Strengths:**

The motivation for the problem setup (within in its assumptions) makes sense for the modern-day end-to-end learned representations.

The math behind the formulations and algorithms checks out and gives us a projection matrix that helps in improving the interpretability and accuracy in certain multi-class settings.

The applications to OOD also make quite a bit of sense.

**Weaknesses:**

I found the paper's fundamental question confusing. If I understand correctly, authors want to take the "spherical" learned embeddings of the images from the penultimate layer -- which often are not disentangled. Then the authors want to compute the class prototype by the class mean which will not be orthogonal to other classes (by design). The goal is to transform these to be orthogonal using techniques like OPL and CIDER and then make them axis-aligned and binary through ICR and DCR. Please correct me if I am wrong in this.

Now the questions are

1) I do not see why we need orthogonal class representations -- because semantically I would like to have a substantial weight in the tail of similar but not the same classes. In case I do not want the semantic similarity between class prototypes to be smooth, one can normalize with the appropriate temperature.

2) The second question is more about why we even need these transformations. If you learn a one-vs-all multi-class classifier for all these data points and classes you will end up generating a "one-hot" vector of dimensionality = number of classes.

This is the regular linear classifier with softmax after all the multi-class image classification networks we learn today. The classifier itself is the projection to ensure you obtain an orthogonal axis-aligned vector for each datapoint and in turn each class. Leaving it at softmax and not thresholing gives your semantically meaningful embeddings that can further be used for class embeddings.

It would be great if these questions could be resolved and the paper heavily leans on OPL and CIDER at times and would be good to have a short background section on the math behind them.

I also point the authors to error-correcting output code line of work (Dietterich & Bakiri, 1995) and probably a concise survey (Kusupati et al 2021) spanning interpretable and learned ECOCs. They aim at learning sub-linear cost class representations in binary space (not necessarily orthogonal, but can be made and also made interpretable in attribute space). This helps result in axis-aligned attribute interpretable binary codes for classes.  This also deals with some notions of OOD detection.



During the rebuttal, if I could get an understanding of this, that would help me be convinced to accept the paper.

**Questions:**

see above

---

> ### Author Response · Authors · 2023-11-17
> **response**
>
> In addition to making class means aligned to axis, we actually make them orthogonal.  Although OPL and CIDR attempted to do this, we show that they came up significantly short of this goal.  Our method resolves this issue in making class means completely orthogonal.
>
>
> Q1.  Why should two classes which are distinct (say apple and orange) be associated?  A fruit cannot be an apple and an orange?  Note that our representation does not stop a new image (not in labeled training set) from have a representation that thinks it might be an apple or an orange.  A data points can have associations with two classes, but the distinct classes do not need correlation with each other.
>
> Q2.  If you just learn classifiers then, either:
>    - these can have the same unnecessary associations discussed in Q1.  If something is clearly an apple, that does not mean a classifier should also guess that it might be an orange.  Note this is indeed what happens without these techniques.
>    - or these classifications are not part of the embedding layer.  So they do not aid to the interpretability.  Our goal is to put as much structure and utility into this one embedding layer.  There are many shifts in industrial machine learning to try to store the results of training as vectors (check out the new company "pinecone").  This aim of this is so that these learned representations capture the main desired properties of the data, and are also more interpretable.
>
> We believe that OPL and CIDR are two examples of ways to initially train a near-orthogonal embedding.  In principal they are somewhat interchangeable (although for SOTA in certain tasks, sometimes one is much better than there others for reasons probably beyond the scope of this paper).  The key part, we believe, is our method which finally achieves full orthogonality, and uncorrelated the representations of distinct labeled classes.
>
> The code book methods seem to have as an aim to compress learned representations.  Our goal is to work within the space of generic vectorized representations, but add enforced structure that do not detract from performance while improving interpretability.

---

> > ### Comment · Reviewer_vbSf · 2023-11-17
> >
> > Thanks for the rebuttal.
> >
> > I get that class prototypes being orthogonal does not mean the images will not have an ordering of the classes based on the inner product. However, it is sample inefficient to pack information that similar classes are as far apart as drastically different classes. I also understand that leveraging them for classification does not affect anything as long as the new basis preserves the orderings.
> >
> > I do not understand you point on the classifiers. A linear layer after embeddings later is still OK with interpretability and the probabilities essentially map to the class prototypes. I am familiar with vector databases, however, I still do not see what this brings to table, over the one hot vector for each classes, when each instance is still a real vector.
> >
> > I think you are missing the point of the codebook methods, your PoV is correct that they might be compressed representations, but they are ideally also solving what you need.
> >
> > I still am not convinced after the rebuttal and willing to discuss further with authors.

---

> > > ### Author Response · Authors · 2023-11-18
> > >
> > > Our thinking is a natural representational goal for learning is a vectorized embedding, so let's see how far we can encode structure and utility into these structures.  Why defer classification into another step (even if it's fairly simple) when it can be directly encoded in the representation.  Ultimately, we think that the vectors in vector databases will be a mix of structured and unstructured parts -- both have advantages.  Our paper makes an argument for this particular balance in structures.
> > >
> > > The codebook approaches are similar, ensuring > Hamming Distance 1 separability between classes, as opposed to orthogonality between classes.  With this they observe additional correlation between related classes, however it does not prohibit unwanted correlation between unwanted classes.  Say for instance, as we do with the example in Figure 3, we want to remove the effect of one of the classes, the binary code representation does not seem to have a mechanism to do this.  There is not a single bit or coordinate to change, and there are k other representations that are 1 bit away, so it's not clear which to move to.
> > > Clearly, their compression is better, but that is not what we are aiming for in this case.

---

### Official Review · Reviewer_E1pC · 2023-10-31

**Soundness:** 2 fair
**Presentation:** 3 good
**Contribution:** 2 fair
**Rating:** 3
**Confidence:** 3

**Summary:**

The paper introduces a post-processing step to the training phase of a learned embedding in the context of  multi-class image classification. The main objective of the step is to obtain orthogonal class means while preserving linearly separable classes at the last layer of the network.
Two algorithms are proposed : Iterative Class Rectification and Discontinuous Class Rectification. Theoretical guarantees of the convergence of both methods are given and numerous experiments are carried out to demonstrate the orthogonality obtained and the preservation of performance in a classification context.

**Strengths:**

- The proposed method is simple, its internal objectives are well described, and the theoretical part seems sound.
- Numerous experiments are carried out

**Weaknesses:**

- The orthogonality objective of the method (although achieved) is not linked to a specific performance improvement in the experimental context,  which makes  its claim somewhat arbitrary. For example, the use of the post-processing step for Out-of-Distribution detection leads to a performance degradation.

- In general, the choice of a smaller ResNet architecture (ResNet-9) does not allow to obtain experimental results equivalent to those provided by the mentioned methods for baseline comparison, although complete results and numerous details are provided.

- The method does not seem to perform consistently depending on the evaluation criteria for image classification. This affects the evaluation of the performance in the case of classification.  Since classification performance does not seem to be the main objective of the post-processing steps : other experiments could have been tried, such as robustness to label noise or robustness to adversarial attacks, as for example in the OPL method paper.

- From a broader point of view, it seems that the goal of orthogonalising the latent space has something in common with  the orthogonal classifier setting, which could provide further experiments and theoretical approaches (See for examples « Controling directions orthogonal to a classifier », ICLR’22).

- As ICR is an extension of ISR (Aboagaye et al.),  the scope of the contribution is limited, despite the scaling capacity of the proposed algorithms.

**Questions:**

-What query related experiments in the SOTA could be used to evaluate the validity of the method ?

---

> ### Author Response · Authors · 2023-11-17
> **response**
>
> The main goals of the proposed methods are:
>   1. Make embeddings more interpretable by having some coordinates associated with class means, and so these aspects are completely orthogonal
>   2. Basically preserve the state of the art.
>
> We achieve these goals.  This method makes no attempt to directly further optimize performance metrics on any tasks, so we should not expect improvement -- although it sometimes happens.

---

> > ### Comment · Reviewer_E1pC · 2023-11-21
> >
> > Thanks for the rebuttal.
> > I think that the equivalence claim between the orthogonality goal and embedding explainability is still insufficiently grounded from a theoretical and experimental perspective.

---

### Official Review · Reviewer_j8xu · 2023-10-31

**Soundness:** 2 fair
**Presentation:** 2 fair
**Contribution:** 2 fair
**Rating:** 3
**Confidence:** 3

**Summary:**

The manuscript presents a mechanism that takes state-of-the-art learned representations and modifies them to assign specific meanings to some of the coordinates. The method makes the representation of each class orthogonal to the others, and then changes the basis to be on coordinate axes. This adjustment aims to improve the interpretability of the representations.

**Strengths:**

- The paper delves into an interesting research direction to  leverage pre-trained encoders
- The authors provide theoretical proofs that underpin the orthogonalization achieved by the proposed methods, granted certain assumptions are met

**Weaknesses:**

### Major weaknesses:

- The experiments, conducted only on a Resnet-9 and limited to CIFAR10 and CIFAR100 datasets, lack the breadth needed to prove the generality of the method.

- The assumption that class means should be entirely orthogonal raises questions. While it makes sense for entirely independent classes, it doesn't account for cases where classes share features or have relationships. The logic behind making every pair of classes orthogonal, especially when some classes naturally have similarities (e.g., apple and orange), remains unclear.

- Despite emphasizing the method's potential for enhanced embedding interpretability, by orthogonalizing them, the paper does not provide empirical evidence that assesses this claim of improved interpretability.

### Weaknesses:
- The method primary objective and results appear straightforward, yet the paper presents the method in a very convoluted manner.

- While the paper suggests possible advantages in downstream tasks by 'ignoring' particular classes or concept, there are not any experiment supporting this claim.

### Minor:
- The paper does not discuss the relationship with prototypical networks, leaving a potentially relevant connection unaddressed.

- The paper claims that "if the representations are successful, then for direct tasks only a simple classifier is required afterwards,", however, it is arguable if a linear layer is always enough after a learned representation
- Typo: Roccio algorithm

**Questions:**

- Why is it necessary for classes to be orthogonal before projecting them into the new basis? Wouldn't it be possible to simply compute the coefficients and execute a change of basis, e.g., using a least squares approach? The rationale behind the choice to orthogonalize vectors first and then execute a change of basis using an orthogonal matrix remains ambiguous. Why not directly learn a non-orthogonal matrix for this transformation?

---

> ### Author Response · Authors · 2023-11-17
> **response**
>
> Weakness #2:   We feel that distinct classes should indeed be uncorrelated.  Although apples and oranges are both fruits they are completely different fruits, and a labeled image is in exactly one such class.  For an unknown object (not labeled in the training set), it is still possible for it to have association with both classes if it is not clear which it is.  However, if it is clearly an apple, that should *not* imply that it is associated with the class orange.  Our orthogonal representation allows this.
>
> Weakness #3:  We believe the example uses in Figure 2,3 provide clear evidence of the interpretability.
>
> Weakness #4:  We have updated the presentation of the method a bit based on another reviewers request.  However, we feel that although the objective is simple to state, it does not have a simple solution (we looked for ones, and prior simpler approaches did not achieve orthogonality).  If you have a simpler approach in mind, that would be great!
>
> Question:  Attempts to learn how to make classes orthogonal (OPL and CIDR) did not fully orthogonalize class means.  We tried other variants which also did not achieve close to full orthogonality.  Our ISR method does.
> What you are describing sounds similar to running one step of ISR; as we show running a few more iterations is required so that the class means of the embedded data are actually orthogonal.  Transformations that orthogonalize the vectors v1 and v2 at the current class means apply to the actual data non-uniformly, so if one recomputes the class means again afterwards, they are no longer orthogonal.

---

> > ### Comment · Reviewer_j8xu · 2023-11-20
> >
> > Thanks for the rebuttal.
> >
> > - (Weakness #2): I am still not convinced that they should be orthogonal, in general. Altough they indeed are completely different fruits, they share many features -- e.g. the shape. An orthogonal representation does not allow to capture shared features, and in general does not allow to learn any meaningful metric between samples.
> >
> > - (Weakness #3): While Figure 2,3 provide examples of interpretability in two specific samples, given the focus on this aspect a deeper qualitative or quantitative analysis would have greatly enhanced the work.
> >
> > I still am not convinced after the rebuttal and willing to discuss further with authors.

---

> > > ### Author Response · Authors · 2023-11-21
> > >
> > > Regarding Weakness #2 and orthogonality:
> > >   Orthogonality could still allow for the embedding network to capture shared features, and if only some shared feature was observed on a test image, then it could be placed to have large dot product along both directions.
> > >
> > > Moreover, imagine a scenario where the training data has numerous pictures of apples and oranges, each labeled correctly.  However, half of each type of image also has a special bit or pattern which accidentally encodes a label (e.g., a logo), and so the classifier can use this to definitely classify them.  Yet, since it's only half of the images, and they share features, then the embeddings left to its own devices will embed the class means or classifier normals for apples and oranges to be aligned.  But then if we get a test data point with the apple logo, so the classifier will definitely know it is an apple, it may be directly aligned with the apple class mean, but also have association with the orange class even though the classifier should know it is not an orange -- it has the apple logo and not the orange logo.  This pattern in representation seems undesirable to us, and the aim of this paper is to address this.  Our orthogonal representations successfully do just that.

---

### Official Review · Reviewer_EbUE · 2023-11-01

**Soundness:** 3 good
**Presentation:** 2 fair
**Contribution:** 3 good
**Rating:** 5
**Confidence:** 4

**Summary:**

While representations in deep learning have become more expressive over time, their interpretability has deteriorated over time as well. This paper advocates to recover the interpretability of the features produced by a deep learning model, by building on top of previous works that encourage sparse and compact representations, and proposing two post-hoc methods that further orthogonalizes these representations. Empirical results show that representations obtained with the proposed methods are indeed orthogonal and axis-aligned, while retaining the classification performance.

**Strengths:**

- The motivation of the work is really well-driven in the introduction section, and it is really appealing.
- Theoretical guarantees for both proposed algorithms are provided.
- Empirical results show the effectiveness of the proposed approaches, as well as the problems of previous approaches.
- I can easily see this work being leveraged by practitioners to improve the interpretability of their features.

**Weaknesses:**

- W1. The introduction of the proposed methods is rather convoluted, short, and unclear. Besides, it relies too much on the reader having previous knowledge of how ISR works. The authors should work on providing more context and explanations to the reader. This is the biggest concern I have with the current state of the manuscript.
- W2. Citations should be fixed, as well as references in the bibliography (e.g., some of them have no venues).
- W3. The explanation of why we cannot simply run Gram-Schmidt (GS) is unclear to me and, in any case, traditional orthogonalization methods (GS, Householder transformations, etc.) should be added as baselines in the experimental section.
- W4. I find the back-and-forth between using OPL vs. CIDER in sections 3.2 and 3.3 a bit too confusing
- W5. The presentation of the results needs a bit more polishing. For example, no statistical results (e.g. standard deviations) are presented, and there is no point in having 3 decimals in Table 4 as the least accuracy is $100/10.000 = 0.01$.

**Questions:**

- Q1. What does it mean that DCR uses a "discontinuous operation"? Discontinuous wrt what exactly?
- Q2. Why is it sensible to normalize the class means? That is, why is it a good idea to completely disregard the magnitude of the class means (since $\arg\min_j D(x, v_j) \neq \arg\min_j D(x, v_j / ||v_j||)$ in general).

---

> ### Author Response · Authors · 2023-11-17
> **response**
>
> Thank you for the feedback on the writing of the paper, we have updated things to try to address these issues.
>
> W3:  Gram-Schmidt can by applied to a pair of vectors (e.g., the class means), but we need an operation that applies to every embedded point, including ones were we may not know the label.  In short, it is not even clear what a baseline of Gram-Schmidt would be.  If there is something we are missing, please let us know what you have in mind.
>
> W4:  We compared against the state-of-the-art in each task.
>
> Q1:  DCR segments the embedding space, and applies a Gram-Schmidt-driven rotation to part of the embedding space, leaving the rest fixed.  The boundary between the transformed space and the fixed space causes a discontinuity in how the update step affects the embedding.
>
> Q2:  The goal is interpretability of the embeddings.  We believe a normalized value will be more interpretable since the length now has some meaning.
> In particular, when the class means are all normalized, then the relative relationship for an evaluation point between classes can be easily understood by directly comparing their coordinates.

---

> > ### Comment · Reviewer_EbUE · 2023-11-22
> >
> > Dear authors, sorry for the late reply, and thanks for your answer. I do not have any further questions. Again, I appreciate the clarifications.

---

### Meta-Review · Area_Chair_u4VK · 2023-12-05

**Metareview:**

The authors propose a way to modify pretrained embeddings so that for a task associated with some set of classes, the new embeddings satisfy a particular notion of disentaglement where the class means are orthogonal. They propose this procedure for the sake of interpretability, quick decoding, bias removal, etc.

The reviewers are largely united in asking for more evidence for the benefits of this procedure. In other words, the debate here is conceptual: is this the right thing to do, rather than do the authors deliver on it? Generally, the authors did not ultimately answer the reviewers' questions, especially around the notion of overlapping or hierarchical classes, where the proposed orthogonal approach doesn't appear to be a good fit. Similarly, it's not clear what will happen in the common case of label noise, etc.

Basically, the paper proposes doing something conceptually interesting that needs more time to be cleared up and fleshed out. The reviewer questions and the discussion are very useful; I believe the next version of the paper will clear the bar.

**Justification For Why Not Higher Score:**

While a fairly interesting proposal, it lacks sufficient justification for how the approach interacts with common classification scenarios. This prevents a higher score.

**Justification For Why Not Lower Score:**

N/A

---

### Decision · Program_Chairs · 2024-01-16

Reject